# The Chalcone Lonchocarpin Inhibits Wnt/β-Catenin Signaling and Suppresses Colorectal Cancer Proliferation

**DOI:** 10.3390/cancers11121968

**Published:** 2019-12-07

**Authors:** Danilo Predes, Luiz F. S. Oliveira, Laís S. S. Ferreira, Lorena A. Maia, João M. A. Delou, Anderson Faletti, Igor Oliveira, Nathalia G. Amado, Alice H. Reis, Carlos A. M. Fraga, Ricardo Kuster, Fabio A. Mendes, Helena L. Borges, Jose G. Abreu

**Affiliations:** 1Program of Cell and Developmental Biology, Institute of Biomedical Sciences, Federal University of Rio de Janeiro, Rio de Janeiro 21941-902, Brazil; 2Department of Chemistry, Federal University of Espírito Santo, Espírito Santo 29075-910, Brazil

**Keywords:** anticancer drugs, flavonoids, natural compounds, *Xenopus laevis*, AOM/DSS model

## Abstract

The deregulation of the Wnt/β-catenin signaling pathway is a central event in colorectal cancer progression, thus a promising target for drug development. Many natural compounds, such as flavonoids, have been described as Wnt/β-catenin inhibitors and consequently modulate important biological processes like inflammation, redox balance, cancer promotion and progress, as well as cancer cell death. In this context, we identified the chalcone lonchocarpin isolated from *Lonchocarpus sericeus* as a Wnt/β-catenin pathway inhibitor, both in vitro and in vivo. Lonchocarpin impairs β-catenin nuclear localization and also inhibits the constitutively active form of TCF4, dnTCF4-VP16. *Xenopus laevis* embryology assays suggest that lonchocarpin acts at the transcriptional level. Additionally, we described lonchocarpin inhibitory effects on cell migration and cell proliferation on HCT116, SW480, and DLD-1 colorectal cancer cell lines, without any detectable effects on the non-tumoral intestinal cell line IEC-6. Moreover, lonchocarpin reduces tumor proliferation on the colorectal cancer AOM/DSS mice model. Taken together, our results support lonchocarpin as a novel Wnt/β-catenin inhibitor compound that impairs colorectal cancer cell growth in vitro and in vivo.

## 1. Introduction

Colorectal cancer (CRC) is the third most commonly diagnosed cancer and the second most common cause of cancer death. According to World Health Organization (WHO), it is expected that there were 1.8 million cases and 862,000 deaths in 2018 [1]. Sporadic CRC initiation, promotion, and progression is mostly driven by a sequence of known genetic mutations in key signaling pathways, frequently related to DNA damage response and sustained proliferation in the absence of growth factors. In CRC, 93% of the cases have at least one mutation of one Wnt/β-catenin pathway component [2]. The most frequently mutated gene in colorectal cancer is *APC* (adenomatous polyposis coli) that is a β-catenin destruction complex component. *APC* mutation occurs in 81% of non-hypermutated colorectal cancers cases and in 51% of hypermutated colorectal cancer cases, triggering tumorigenesis in intestinal polyps of patients with familial adenomatous polyposis [3]. The Wnt/β-catenin signaling pathway coordinates several cell behavior aspects, such as cell proliferation, differentiation, stemness, polarity, and migration [4,5]. In the absence of Wnt ligands, the destruction complex is active in the cytoplasm, phosphorylating β-catenin, a key component of the canonical Wnt pathway, leading to its degradation by the proteasome [6,7]. Wnt interaction with its receptors Frizzled (Fzd) and LDL receptor-related protein 5/6 (LRP5/6) disrupts the destruction complex assembly leading to β-catenin stabilization, cytoplasmic accumulation, translocation to the nucleus and binding to the T-cell factor/lymphoid enhancer factor (TCF/LEF), allowing Wnt target gene transcription [8]. Despite the crucial role of Wnt signaling on colorectal tumorigenesis, there is no Wnt/β-catenin inhibitor approved for clinical use [9]. Due to the importance of Wnt/β-catenin and its frequent mutations upstream to β-catenin translocation to the nucleus, it is crucial to find anticancer drugs that target the pathway downstream to this phenomenon [2].

Addressing normal and pathological Wnt/β-catenin signaling functioning requires multidisciplinary experiments combining in vitro and in vivo approaches. Among different models for studying Wnt/β-catenin signaling in vivo, *Xenopus laevis* stands out for its liability and efficiency. Wnt/β-catenin signaling plays a key role in two fundamental steps during the Xenopus early development that can be exploited for the screening of new drug candidates: the dorso-ventral and the antero-posterior axis patterning [10,11,12]. Indeed, the Xenopus model system has been explored to discover Pyrvinium, an FDA approved compound, as a Wnt signaling inhibitor that acts downstream of β-catenin. Pyrvinium impaired Xenopus embryo secondary axis induction in a dose-dependent manner and decreased colon cancer cells viability [13].

In addition, the AOM/DSS mouse model stands as a relevant preclinical inflammation-associated CRC model with histologic and phenotypic features that recapitulates the aberrant crypt foci-adenoma-carcinoma found in the human CRC [14]. Consistent with CRC development, in the AOM/DSS murine model, β-catenin nuclear translocation is observed in both flat and polypoid lesions likely due to β-catenin mutation [15]. In this context, the study of synthetic and natural compounds able to inhibit the Wnt/β-catenin signaling pathway have been explored as possible antitumor prototypes. Among the small natural molecules studied, the flavonoids, polyphenolic compounds found in many plants with a wide range of biological effects, stand out. Many flavonoids have been described as inhibitors of Wnt signaling and potential antitumor compounds, such as apigenin, EGCG, silibin, kaempferol, isorhamnetin, quercetin, isoquercitrin, derricin, and derricidin [16,17,18,19,20,21,22,23,24,25]. However, the specific mechanism by which some of these compounds affect Wnt/β-catenin signaling as well as its capacity to impair CRC growth is still not elucidated. Along the flavonoid biosynthesis pathway, the chalcones are well known as precursors of the flavonoids. Lonchocarpin is a chalcone first isolated from *Lonchocarpus sericeus* (as known as *Derris sericeu*) by Baudrenghien and colleagues in 1949 [26], and its correct structure was elucidated by the same researchers in 1953. The cytotoxic effects of the chalcone lonchocarpin have been previously described in neuroblastoma and leukemia cell lines [27], however, its role in CRC and the Wnt/β-catenin pathway is unknown.

In the present work, we describe lonchocarpin ability of inhibiting Wnt/β-catenin both in vitro, in colon cancer cell lines, and in vivo, with Wnt specific *Xenopus laevis* embryonic assays. In addition, acute administration of lonchocarpin in a preclinical CRC mouse model reduced cell proliferation in adenocarcinomas. Altogether, our data show lonchocarpin as a potent Wnt/β-catenin inhibitor that impairs cancer cell proliferation both in vitro and in vivo, and a promising compound for further antitumor clinical investigation and development.

## 2. Results

### 2.1. Lonchocarpin Inhibits Wnt/β-Catenin Pathway and Reduces Nuclear β-Catenin Levels

It has been shown that natural compounds, including chalcones, have growth-inhibitory properties in cancer cell lines by modulating Wnt/β-catenin signaling [17,18]. We employed an RKO pBAR/Renilla based screening of natural compounds and found lonchocarpin as a Wnt signaling modulator hit (data not shown). In this context, we evaluated whether lonchocarpin, a chalcone isolated from *Lonchocarpus sericeus*, is able to inhibit the Wnt signaling pathway in human colorectal cancer cell lines RKO pBAR/Renilla stimulated with Wnt3a CM (conditioned medium) for 24 h and non-stimulated SW480 pBAR/Renilla (Figure 1A). SW480 does not need Wnt stimulation, since it harbors an APC mutation that overactivates the canonical Wnt signaling. Cells were treated with lonchocarpin overnight, with or without the conditioned medium. Lonchocarpin decreased luciferase reporter activity in a concentration-dependent manner, starting at 3 µM in RKO pBAR/Renilla and 5 µM in SW480 pBAR/Renilla (Figure 1B,C), and presented the half maximal inhibitory concentration (IC50) of 4 µM in SW480 pBAR/Renilla (Figure 1D).

In order to validate the Wnt/β-catenin reporter gene assay inhibition, we performed immunocytochemistry on SW480 cells. In order to activate the pathway, we treated RKO cells with Wnt3a CM for 24 h and performed immunocytochemistry to assess nuclear β-catenin cell count. These cells were cotreated with DMSO (vehicle), 10 or 20 µM lonchocarpin. L-cell CM treated cells displayed 34% β-catenin positive nuclei (Figure 1E–E”), while Wnt3a CM treatment increased the mean to 86% (Figure 1F–F”). RKO cells cotreated with 10 µM (Figure 1G–G”) or 20 µM Figure 1H–H”) lonchocarpin showed 54% and 30% nuclear β-catenin positive cells, respectively (Figure 1J). This data show that lonchocarpin treatment decreased β-catenin nuclear translocation in a concentration-dependent manner (Figure 1E–H”). To assess whether lonchocarpin affects total β-catenin protein level, we performed immunoblotting assay of RKO cells stimulated overnight with Wnt3a CM or 1 µM BIO, cotreated with DMSO or 20 µM lonchocarpin (Figure 1K). Immunoblot assay showed that lonchocarpin did not expressively reduced β-catenin total levels (Figure 1K), suggesting that β-catenin degradation, or β-catenin stabilization impairment might not be a lonchocarpin mechanism of action. In order to further validate our immunocytochemistry analysis, we performed immunoblotting of RKO cytosolic and nuclear fractions (Figure 1L). These cells were treated accordingly to previous assay. Immunoblotting showed that lonchocarpin reduces β-catenin nuclear level, suggesting an inhibition of nuclear translocation, or, possibly, an inhibition of β-catenin nuclear interaction with other proteins or DNA. Thus, this data suggests that lonchocarpin inhibits Wnt/β-catenin signaling pathway and impairs β-catenin nuclear localization.

### 2.2. Lonchocarpin Inhibits the Canonical Wnt Pathway Downstream of the Destruction Complex

SW480 harbors a mutation in APC that deletes its carboxyl-terminus domain, preventing the destruction complex assembly [28,29]. Considering that lonchocarpin inhibited the Wnt reporter gene in SW480 pBAR/Renilla (Figure 1C,D), and also prevented β-catenin nuclear localization (Figure 1L), we speculated that the flavonoid could act downstream of the destruction complex. To test this hypothesis, we performed epistasis assay using Wnt/β-catenin-specific reporter (Super TOPFLASH) transfected cells induced by Wnt3a CM (Figure 2A), or co-transfected with wild type β-catenin (Figure 2B), constitutively active β-catenin S33A (Figure 2C), or dnTCF4 VP16 (Figure 2D), a constitutive active TCF4 form that does not rely on β-catenin for inducing the pathway.

Our results showed that lonchocarpin inhibits Wnt/β-catenin signaling activation when pathway activation is triggered by wild-type β-catenin, β-catenin S33A, or dnTCF4 VP16 overexpression (Figure 2A–D). Interestingly, lonchocarpin showed a more potent and efficient inhibitory effect than the previously published compound PNU-74654 (Figure 2A–D) [30]. In this scenario, we propose that lonchocarpin acts downstream of the destruction complex and impairs TCF4 mediated transcription.

Together, these data suggest that lonchocarpin inhibits Wnt/β-catenin signaling by impairing β-catenin nuclear localization while also hindering TCF activity.

### 2.3. Lonchocarpin Treatment Disturbs Xenopus laevis Embryos Axial Patterning and Rescues Wnt Overactivation Phenotypes

Our in vitro data suggest that lonchocarpin inhibits Wnt/β-catenin by impairing β-catenin nuclear localization (Figure 1E–L) while also suppressing TCF mediated transcription (Figure 2D). Next, we investigated if this inhibitory effect is also observed in vivo. Manipulation of *Xenopus laevis* embryonic development has been successfully used to validate compounds targeting Wnt/β-catenin signaling through the interpretation of embryonic phenotypes and Wnt signaling overactivation [11]. In this context, injection of 1.6 pmol lonchocarpin in the embryo animal dorsal blastomeres (Figure 3A) induced head defects, characterized by reduction of anterior structures in 23% of injected embryos, such as the cement gland and diminished eyes (Figure 3D) while uninjected or DMSO-injected embryos developed normally (Figure 3B,C). Considering that Wnt/β-catenin is active in the dorsal side of the Xenopus embryo, which will organize the anterior-posterior axis, this result suggests an inhibition of the pathway by lonchocarpin. In order to confirm this result, we injected 10 pmol of lonchocarpin into the blastocoele space of stage 9 embryos (Figure 3E). Lonchocarpin injection induced anterior structures enlargement, such as the head and cement gland in 48% of the embryos (Figure 3H), while uninjected and DMSO injected embryos developed normally (Figure 3F,G). This lonchocarpin effect is consistent with Wnt/β-catenin inhibition in the embryo anterior region, since the signaling at this embryonic stage induces posterior structures.

It is well established that Wnt/β-catenin signaling overactivation in the Xenopus embryo 4-cell stage ventral side induces ectopic axis containing head and dorsal structures [10,12,31]. We injected 10 pg of xWnt8 mRNA combined or not with 10 pmol lonchocarpin or DMSO into two ventral blastomeres at 4-cell stage embryo (Figure 3I). xWnt8 or xWnt8 + DMSO-injected embryos developed a secondary axis in 90% of the embryos, while only 60% of the lonchocarpin-injected embryos developed a secondary axis (Figure 3K–M,P).

Thus, these results strongly suggest lonchocarpin induces phenotypes consistent with Wnt/β-catenin inhibition in Xenopus embryos (see also Appendix A for quantification). To confirm whether lonchocarpin inhibits Wnt signaling in Xenopus, we coinjected a Wnt/β-catenin specific gene reporter S01234-luciferase with xWnt8 mRNA and 2.4 pmol lonchocarpin or DMSO into 4-cell *stage Xenopus laevis* embryos (Figure 3N). Lonchocarpin suppressed 82% of Wnt/β-catenin signaling gene reporter activation (Figure 3O).

These data support that lonchocarpin inhibits Wnt/β-catenin signaling pathway activation both in vivo and in vitro.

### 2.4. Lonchocarpin Reduces HCT116, SW480, and DLD-1 Cell Proliferation

Since lonchocarpin inhibited Wnt/β-catenin downstream to the destruction complex both in vitro and in vivo, we asked whether lonchocarpin has antitumor effects on colorectal cancer cell lines where Wnt/β-catenin signaling has a critical role on tumorigenesis. Canonical Wnt signaling activation leads to proliferation in many cell types, including colorectal cancer cells, thus we asked whether lonchocarpin inhibits colon cancer cell proliferation. We treated HCT116, SW480, DLD-1, and IEC-6 cell lines with 5, 10, or 20 µM lonchocarpin for 24 h and performed the Click-it EdU proliferation assay.

Lonchocarpin inhibited 33% of HCT116 EdU positive cell count at 5 and 10 µM and reduced 75% of EdU positive cell count at 30 µM (Figure 4B–E). Lonchocarpin inhibited 50% and 85% of SW480 EdU positive cell count at 10 and 20 µM, respectively (Figure 4H–J). Lonchocarpin inhibited 40% and 75% of DLD-1 proliferative cells ratio at 10 and 20 µM, respectively (Figure 4M–O). However, lonchocarpin did not affect the non-tumoral intestinal cell line IEC-6 proliferating cells ratio (Figure 4Q–T). These data show that lonchocarpin suppresses colorectal cancer cell lines proliferation while not affecting the non-tumoral cell line IEC-6 proliferation. Next, we asked whether lonchocarpin suppresses proliferation through cell toxicity. In order to assess cellular viability, we performed MTT assay after treatment with 10, 20, 30, 40, and 50 µM lonchocarpin for 24, 48, or 72 h (Figure 4). We noticed that 20 to 50 µM lonchocarpin at all treatment intervals reduced relative 570 nm absorbance of HCT116, SW480, and DLD-1 colorectal cancer cell lines but 10 µM had no effect (Figure 4U–W). Curiously, 20 µM lonchocarpin did not decrease relative 570 nm absorbance, but maintained the same reading throughout the experiment, suggesting a proliferation inhibition.

However, in the non-tumoral intestinal cell line IEC-6, lonchocarpin decreased relative 570 nm absorbance only at 50 µM. Exclusively at 72 h of treatment there was noticeable absorbance reduction in 40 µM lonchocarpin (Figure 4X). These data show that lonchocarpin inhibits cell proliferation and viability preferentially in the colorectal cancer cell lines.

### 2.5. Lonchocarpin Reduces Cell Migration in HCT116, SW480, and DLD-1 Colorectal Cancer Cell Lines

Canonical Wnt signaling key protein β-catenin interacts with adhesion proteins in the membrane that may affect cell adhesion and migration. We evaluated whether lonchocarpin affects HCT116, SW480 and DLD-1 colorectal cancer cell lines and IEC-6 non-tumoral intestinal cell line migration by performing scratch assay. HCT116 lonchocarpin treatment reduced 55% of scratch closure at 20 µM (Figure 5A–E). Lonchocarpin treatment of SW480 reduced 40% and 55% of scratch closure at 10 and 20 µM, respectively (Figure 5F–J). Likewise, DLD-1 lonchocarpin treatment reduced 45% of scratch closure at 20 µM (Figure 5K–O). However, lonchocarpin treatment did not affect cell migration of the non-tumoral cell line IEC-6 (Figure 5P–T). These data show that lonchocarpin treatment impairs cell migration of colorectal cell lines, while not affecting the migration of the non-tumoral cell line.

### 2.6. Lonchocarpin Decreases Cell Proliferation in Azoxymethane (AOM)/Dextran Sulfate Sodium (DSS) Induced Adenocarcinomas

Lonchocarpin inhibits Wnt/β-catenin in vitro and in vivo, while also presenting antitumor effects in vitro. We asked whether lonchocarpin also has antitumor effects in vivo. We assessed the efficacy of lonchocarpin therapeutic administration in an azoxymethane (AOM)/dextran sulfate sodium (DSS)-induced model of colon cancer. After three cycles of DSS, when colon tumors were expected in most animals, lonchocarpin (50 or 100 mg·kg^−1^·day^−1^) was injected intraperitoneally for four days, and the mice were assessed 3 h after the last injection (Figure 6A). Colon tumors were macroscopically observed in almost all mice submitted to the protocol (77%), and histopathological analyses revealed several changes in the intestinal mucosa (Appendix A). The most frequent alterations included no presence of mononuclear and polymorphonuclear leukocyte infiltrates in the lamina propria and submucosa, hyperplastic epithelium, in addition several adenomas and adenocarcinomas (Figure 6B). Lonchocarpin did not show any toxicity to the treated animals, but significantly reduced tumor proliferation (Figure 6C–F”). Lonchocarpin at 100 mg·kg^−1^·day^−1^ significantly decreased 31% and 38% of proliferative Ki-67 and BrdU positive cells in adenocarcinomas of the treated mice compared either with vehicle groups, respectively (Figure 6G,H, Appendix A). However, lonchocarpin showed a more efficient antiproliferative effect when administered at the highest dosage (100 mg·kg^−1^·day^−1^) in comparison to the lower dosage (50 mg·kg^−1^·day^−1^), in which no statistical relevance was found (Figure 6G,H, Appendix A).

Taken together, our data demonstrate that lonchocarpin suppress the colorectal cancer cell growth in vitro and in vivo.

## 3. Discussion

The Wnt/β-catenin pathway plays a key role in colorectal tumorigenesis Integrated analysis of 195 colorectal tumors revealed that Wnt signaling pathway components were mutated in 94% of all tumors, and these mutations occurred mostly downstream to APC^10^. Hence, describing the novel canonical Wnt signaling pathway inhibiting small molecules that act downstream to APC is a recurrent strategy to improve colorectal cancer treatability.

Several studies show the antitumor effect of natural compounds that act as inhibitors of multiple components of the Wnt/β-catenin pathway [17]. Quercetin has been described to disrupt TCF/β-catenin interaction [23,32]. Epigallocatechin-3-gallate (EGCG) has been shown as a Wnt/β-catenin inhibitor by promoting β-catenin degradation [33]. Isoquercitrin has been described as an inhibitor of the Wnt signaling both in vitro and in vivo, and impairing tumor growth in vitro [34]. The chalcones derricin and derricidin have also been reported as canonical Wnt signaling inhibitors impairing tumor growth in vitro [18].

The present work is the first to identify lonchocarpin as a negative modulator of the Wnt/β-catenin pathway in RKO and SW480 colon tumor cell lines and in the HEK293T embryonic kidney cell line. We further elucidate that lonchocarpin acts downstream to β-catenin stabilization, probably at the TCF level, since it inhibits the overactivation of the Wnt/β-catenin pathway induced by the transfection of wild-type β-catenin, constitutively active β-catenin S33A or the constitutively active dnTCF4 VP16 in the HEK293T epistasis assay, while also inhibiting the Wnt/β-catenin pathway reporter in SW480 pBAR/Renilla cells, which harbors an APC truncation. We also show that lonchocarpin inhibits proliferation, migration, and cell viability in most of the three colorectal cancer cell lines, HCT116, SW480, and DLD-1, while not altering any of these aspects of the non-tumoral intestinal rat cell line IEC-6.

Recent work has identified that 24 h treatment with 50 μM lonchocarpin of SK-N-SH neuroblastoma line induces AMPK phosphorylation, which results in increased glucose uptake and inhibits protein synthesis [27]. Although our data shows effects at lower concentrations, this work corroborates our findings that lonchocarpin decreases cell viability. Thus, the antitumor effects that we described may be also a consequence of modulation of other signaling pathways besides Wnt/β-catenin.

Previous work has also measured cell viability following lonchocarpin treatment through MTT assay, indicating that the IC_50_ for cell growth in the CEM leukemia cell line is 10.4 µg/mL, the same as 3.4 µM of lonchocarpin [35]. Although not explored by the authors, leukemia cell lines are known to have Wnt signaling activating mutations, and lonchocarpin growth inhibiting effect could be due to Wnt signaling inhibition [36]. In this same study, authors show that derricin also inhibits leukemia cell growth [35], a chalcone also described as a canonical Wnt signaling inhibitor [18]. Comparatively, our cell viability data in HCT116, SW480, and DLD-1 colorectal cancer cell lines show that lonchocarpin reduces cell viability starting at 20 μM. This disparity may be due to the different origin of the cell lines. Lonchocarpin has also been shown as inhibiting lung cancer cells H292 growth in vitro and murine sarcoma S180 graft growth in vivo by inducing Caspase-3 mediated cell death [37]. Curiously, increased cleaved Caspase-3 levels were found at 48 h [37], but not at 24 h, suggesting that canonical Wnt signaling is inhibited prior apoptosis induction. Thus, previous lonchocarpin biological effect descriptions confirm that the antitumor effect of this chalcone is not exclusive to colorectal cancer.

Additionally, we show that lonchocarpin inhibits Wnt/β-catenin signaling in vivo in the *Xenopus laevis* embryo model. Xenopus embryo is a robust and reliable system to approach Wnt/β-catenin signaling in vivo since it is critical for axis patterning [10]. In that sense, Wnt signaling plays two major roles in early Xenopus development: a prior role that is composed by maternal Wnt components, and a latter role that is regulated by genes transcribed by the embryo itself. We show that injection of lonchocarpin at 4-cell stage, a stage where the embryo has no functioning transcription machinery, induced microcephaly and impaired Wnt8-induced axis, although a not very strong effect, most likely due to low lonchocarpin concentration into the embryo blastomere (Figure 3). Consistently, lonchocarpin injection into the blastocoele as well as co-injection with Wnt8-specific reporter (S01234), both addressing a moment where the embryo transcribes genes by itself, induced head enlargement and suppressed 82% of Wnt/β-catenin signaling gene reporter activation (Figure 3). These results are consistent with previous epistasis assay showing that lonchocarpin inhibits Wnt pathway downstream to β-catenin stabilization level, by impairing TCF mediated transcription (Figure 2).

Considering the relevance of the lonchocarpin Wnt/β-catenin inhibition and its functional effects on CRC cell lines as well as in Xenopus embryo, we tested lonchocarpin therapeutic administration in an inflammation-associated CRC mouse model. The AOM/DSS model has been widely used for CRC studies since it is highly reproducible, and recapitulates human cancer histological features and the major driven mutations. Indeed, it has become a powerful platform for chemopreventive and anticancer drug discovery studies. AOM is a procarcinogen metabolized into alkylating agent methylazocymethanol (MAM) in the liver. After excretion into the bile, it induces mutagenesis of the colonic epithelium. Colonic tumors are accelerated by a heparin-like polysaccharide DSS, which causes colonic epithelial damage, mirroring some of the features of inflammatory bowel disease [38]. Acute lonchocarpin i.p. administration in mice containing fully developed carcinomas reduced Ki-67 positive and BrdU positive cell count (Figure 6). Therefore, indicating lonchocarpin as an acute anti-proliferative agent in AOM/DSS induced adenocarcinoma. Although very promising, only the lonchocarpin highest dose produced anti-proliferative effects suggesting that its biological activity should be further enhanced through the synthesis of novel optimized analogues, or by using alternative administration approaches. In the middle of the 1950s, it was first isolated from the stem wood of *Camptotheca acuminata* the precursor of one of the most used chemotherapeutic agents for colorectal cancer treatment, the alkaloid camptothecin [39]. First described as showing antileukemic and antitumor activities, for the following three decades many camptothecin analogues have been described in order to enhance its antitumor properties. One of these analogues was CPT-11, currently known as Irinotecan, that is widely used to treat colorectal cancer [40]. Together with Irinotecan, 5-FU is also widely used clinically. Intriguingly, Wnt signaling inhibition has been shown to decrease resistance of colorectal cells to these chemotherapy drugs’ treatment [41]. Interestingly, MEK signaling inhibitors have been shown to increase canonical Wnt signaling in CRC [42], and the co-treatment of MEK inhibitors and Wnt signaling inhibitors resulted in a reduction of tumor growth [42]. Thus, MEK inhibitors should also be addressed as a cotreatment with lonchocarpin.

There are currently 55 clinical trials aiming to inhibit Wnt signaling pathway in cancers, in which 21 are CRC studies (clinicaltrials.gov). Among the 21 CRC clinical trials, only two tested compounds inhibit canonical Wnt signaling at the transcriptional level, PRI-724 and resveratrol. In this context, in comparison with these compounds, the in vitro IC50 of lonchocarpin Wnt signaling inhibition is noticeable. PRI-724 inhibited the Wnt signaling pathway at 25 μM in vitro [43] (authors used cyclin D1 Western blot to check Wnt signaling modulation), resveratrol inhibited at 20 μM [44] (authors used a Wnt signaling specific gene reporter assay), whereas 4 uM lonchocarpin reached the IC50. The PRI-724 clinical trial has been withdrawn due to supply issues (NCT02413853) and resveratrol results have not been published yet (NCT00256334). We believe that lonchocarpin anticancer effects should be further addressed in preclinical studies, so it can be a possible clinical trial candidate.

The use of natural compounds as drug candidates has been improved by the use of alternative delivery approaches such as controlled delivery systems. Indeed, nanostructuration has been used to circumvent solubility issues [45,46,47]. These strategies can solve instability and poor water solubility issues [48,49] and improve a drug candidate pharmacokinetic profile. These systems could be used to further enhance lonchocarpin anticancer properties in vivo.

At last, the similarity of lonchocarpin and derricin chemical structure deserves to be noticed. These two natural compounds inhibit Wnt signaling through similar mechanisms, while derricidin has a different one [18]. This comparison paves a new way for structure-function studies, and the quest for new Wnt signaling inhibitor pharmacophores.

## 4. Materials and Methods

### 4.1. Cell Lines and Chemical Compounds

HEK293T, SW480, HCT116, DLD-1, and IEC-6 cell lines were purchased from ATCC and RKO-pBAR/Renilla and SW480-pBAR/Renilla were a gift from Professor Xi He (Harvard Medical School). All cell lines were maintained in DMEM-F-12 (Gibco, Life Technologies Limited, Paisley, UK) enriched with 10% fetal bovine serum (Gibco). The chalcone lonchocarpin was purified by Nascimento and Mors, 1972 [50] and kindly donated for this study by professor Ricardo Kuster (Federal University of Espirito Santo). PNU-74654 was synthesized at the Laboratory of Evaluation and Synthesis of Bioactives substances (Biomedical Sciences Institute, UFRJ). Both compounds were diluted in DMSO (Sigma-Aldrich, Saint Louis, MO, USA) at the concentration of 10 mM. PNU-74654 has been previously described as an inhibitor of Wnt/β-catenin pathway by blocking the interaction between β-catenin and TCF [30]. L-cell conditioned medium (CM) and Wnt3a CM were obtained according to the ATCC protocol. L-cells and L-Wnt3a cells were plated into 75 cm^2^ flasks at 50% confluence with 10 mL DMEM medium containing 10% FBS. After 4 days, the first batch of medium was obtained and kept at 4 °C. Three days later, the last batch of medium was obtained and combined with the first one. Finally, L-cell CM and Wnt3a CM were passed through a 0.22 μm filter, and kept at 20 °C.

### 4.2. Wnt/β-Catenin Luciferase Reporter Assay

First, 1.2 × 10^4^ cells/well RKO-pBAR/Renilla and SW480-pBAR/Renilla cells were cultured on 96-well plates in DMEM/F-12 containing 10% fetal bovine serum (Gibco). Then, 24 h later, cells were treated with lonchocarpin at the concentrations of 1, 3, 5, 10, 20, and 30 µM in the presence of Wnt3a conditioned medium for an additional 24 h. L-cell CM was used as negative control, and 0.3% DMSO was used as vehicle control. After 24 h of treatment, Firefly and Renilla luciferase activities were detected according to the manufacturer’s protocol (Dual Luciferase Reporter Assay System, Promega, Madison, WI, USA).

The 1.2 × 10^4^ cells/well HEK293T cells were cultured on 96-well plates in DMEM/F-12 containing 10% fetal bovine serum (Gibco). At 70% confluence, each well was transfected with 100 ng TOP-Flash plasmid, 10 ng Renilla-luciferase plasmid, and 100 ng wild type β-catenin or 100 ng β-catenin S33A using Lipofectamine 3000 (Invitrogen, Life Technologies Corporation, Carlsbad, CA, USA). Then, 18 h after transfection, cells were treated with 1, 5, and 10 µM of lonchocarpin. After 24 h, Firefly and Renilla luciferase activities were detected according to the manufacturer’s protocol (Dual Luciferase Reporter Assay System, Promega).

### 4.3. Immunocytochemistry

SW480 cells were cultured in 24-well plates with 4.0 × 10^4^ cells/well in DMEM-F12 media (Gibco) containing 10% fetal bovine serum (Gibco). Cells were fixed in 4% paraformaldehyde, washed with phosphate buffered saline, and permeabilized with 0.3% Triton X-100. Then, each well was blocked for 1 h with 5% bovine serum albumin. The rabbit anti-β-catenin primary antibody (Sigma Aldrich) was diluted in PBS containing 1% bovine serum albumin (1:200) and incubated overnight. The secondary antibody anti-rabbit Alexa Fluor 546 (Sigma Aldrich) was diluted (1:500) with 1% bovine serum albumin (1:500) and incubated for 2 h at room temperature. 4,6-diamidino-2-phenylindole staining (Cell Signaling) was performed for 15 min, and then slides were mounted with Fluoromount (Sigma Aldrich). Images were captured using the confocal microscope Leica TCS SP5.

### 4.4. Immunoblotting and Cell Fractionation

First, 2 × 10^5^ cells/well RKO-pBAR/Renilla cells were cultured on 12-well plates and treated with 20 μM lonchocarpin for 24 h. Then, the cells were harvested in ice-cold PBS 1X, followed by cytosolic lysis in digitonin buffer with protease inhibitors (150 mM NaCl, 50 mM Tris, 25 μg/mL digitonin, pH 7.4) for 5 min. Then, the lysate was centrifugated for 10 min at 2000 G at 4 °C and the supernatant was collected and considered as the cytosolic fraction. Following digitonin extraction, the remaining cell pellets were washed in ice-cold PBS 1X and lysed in NP-40 buffer (150 mM NaCl, 50 mM Tris, 1%NP-40, pH 7.4) with protease inhibitors for 15 min and centrifuged for 10 min at 7000 G at 4 °C. The supernatant was collected and considered as the membrane fraction. After that, the cell pellets were washed again in ice-cold PBS 1X and lysed in RIPA buffer with protease inhibitors for 20 min and centrifugated after at 16,000 G at 4 °C. The supernatant was collected and considered as the nuclear fraction. Whole cell lysates were prepared using Triton buffer (150 mM NaCl, 50 mM Tris, 1% Triton X-100, 1 mM EDTA, 10% Glycerol, pH 7.5). Finally, cell lysates were denatured with Laemmli buffer at 95 for 5 min, and the protein samples were subjected to SDS-PAGE and transferred into PVDF membranes (Millipore, Merck KGaA, Darmstadt, Germany). The membrane was then blocked with 2% PVP (Sigma Aldrich) in TBS-Tween-20, incubated with primary monoclonal antibodies β-catenin (BD, 1:500), β-actin (SCBT, 1:2000); lamin A/C (CST, 1:500) and α-tubulin (Sigma, 1:2000) overnight at 4 °C. After three washes with TBS-T, the membranes were incubated for 1 h with HRP-conjugated secondary antibodies (CST). The immunoblots were visualized by chemiluminescence using SuperSignal West Pico and West Femto (ThermoFisher, Life Technologies Corporation, Carlsbad, CA, USA).

### 4.5. Xenopus laevis Embryo Manipulations

Frog experiments were carried out according to the guidelines granted by the Animal Care and Use Ethic Committee from the Federal University of Rio de Janeiro and were approved by this committee under the permission number 152/13. Female adult frogs (Nasco Inc., Fort Atkinson, WI, USA) were stimulated with 1000 IU human chorionic gonadotropin (Ferring Pharmaceuticals, Kiel, Germany). *Xenopus laevis* embryos were obtained through in vitro fertilization and staged according to Nieuwkoop and Farber [51]. All experiments were performed at 22 °C. For the Wnt/β-catenin signaling specific reporter assay, two transverse blastomeres of 4-cell stage embryos were injected with 4 nL containing 280 pg of S01234-luciferase plasmid, 50 pg of TK-Renilla plasmid, 1 pg of xWnt8 mRNA, and 1.2 pmol of lonchocarpin or DMSO each, for a total of 2.4 pmol of lonchocarpin per embryo. For synthetic xWnt8 mRNA, the plasmid was linearized with NotI and transcribed with SP6 RNA polymerase using the mMessage mMachine kit (Applied Biosystems, Austin, TX, USA). After microinjections, embryos were cultivated in 0.1× Barth (8.89 mM NaCl; 0.1 mM KCl; 0.24 mM NaHCO_3_; 0.08 mM MgSO_4_·7H_2_O; 1 mM Hepes; 0.03 mM Ca(NO_3_)_2_·4H_2_O; 0.04 mM CaCl_2_·2H_2_O; pH 7.7) until sibling control embryos reached stage 10. Triplicates of four embryos were lysed using 50 µL of 1× Passive Lysis Buffer (Promega). Then, 10 µL of embryo lysate was used for gene reporter activity detection. Firefly and Renilla luciferase activities were detected according to the manufacturer’s protocol (Dual Luciferase Reporter Assay System, Promega).

In order to modulate the maternal wave, 4-cell stage embryos dorsal blastomeres were injected in the animal pole with 4 nL containing 200 µM of lonchocarpin (0.8 pmol/embryo) or DMSO each, for a total of 1.6 pmol of lonchocarpin per embryo. In order to modulate the zygotic wave, stage 9 embryos were injected with 50 nL containing 200 µM of lonchocarpin (10 pmol/embryo) or DMSO into the blastocoel. After injection, embryos were maintained in 0.1X Barth, until stage 35, when the phenotypes were analyzed.

### 4.6. MTT Assay

3-(4,5-Dimethylthiazol-2-yl)-2,5-diphenyltetrazolium bromide (MTT) was used to assay mitochondrial activity in viable cells. Cells were plated at a concentration of 1.2 × 10^4^ cells/well in 96-well tissue culture plates in DMEM/F-12 containing 10% fetal bovine serum (Gibco) and cultured for 24 h. Cells were treated for 24 h with 10, 20, 30, 40, or 50 µM of lonchocarpin and 0.5% DMSO was used as the vehicle control. MTT was added to each well at a final concentration of 0.25 mg/mL for 1 h. The formazan reaction product was dissolved with 100% DMSO and quantified spectrophotometrically at 570 nm (Modulus™ II Microplate Multimode Reader, Promega).

### 4.7. Scratch Assay

HCT116, SW480, DLD-1, and IEC-6 were cultured on 12-well plates in DMEM/F-12 containing 10% fetal bovine serum (Gibco). Confluent wells were scratched in the center of each well with a pipette tip and treated with the proliferation inhibitor Ara-C at 10 µM. After scratch, cells were treated with 5, 10, or 20 μM lonchocarpin for 24 h. Images were taken at 0 h and the wound areas were measured at 0 and 24 h after treatment. Each experiment was carried out in triplicate, and three fields were counted per well.

### 4.8. Cell Proliferation Assay

HCT116, SW480, DLD-1, and IEC-6 were cultured on 24-well plates with 4.0 × 10^4^ cells/well in DMEM/F-12 containing 10% fetal bovine serum (Gibco). Then, 24 h later, cells were treated with 20 or 30 μM lonchocarpin. DMSO 0.3% was used as vehicle control. Then, 18 h later, we added EdU to the cells, and 6 h later cells were fixed with paraformaldehyde 4% and the experiment was conducted according to Click-iT EdU (Life Technologies Corporation, Carlsbad, CA, USA) manufacturer protocol.

### 4.9. AOM/DSS Protocol

Animal procedures were approved and carried out according to the guidelines by the Animal Care and Use Ethic Committee from the Federal University of Rio de Janeiro under register 85/15. Male and female 129SvJxC57BL6 mixed mice (8–12 weeks) were housed in microisolator cages and maintained at 23 °C with a 12/12-h light/dark cycle and free access to food and water. A total of 26 experimental mice were pre-treated with vermifuge (vetmax plus 0.04% and ivermectin 0.4%) in quarantine for 5 days. All mice were intraperitoneally injected with AOM (12.5 mg/kg; Sigma-Aldrich) once. Five days later, all animals received the first cycle of DSS treatment composed by 2% DSS (MP Biochemicals, Solon, OH, USA) in drinking water for five days, followed by a 2-week rest period without DSS. The 2% DSS treatment cycle was repeated once and followed by a last cycle of 1.5% DSS for 4 days. Mice were monitored every day. Any mouse that lost greater than 20% body weight, demonstrate hunched posture, or moved in a limited fashion was euthanized. Four weeks after the last DSS cycle the animals were randomized into five groups as follows: no treatment group (only AOM/DSS); i.p. injections of 50 or 100 mg·kg^−1^·day^−1^, of vehicle (30% polyethylene glycol 400 (PEG400) with 0.9% saline); and i.p. injections of 2.5 mg/mL lonchocarpin 50 or 100 mg·kg^−1^·day^−1^ for four days. Due to limitation of the lonchocarpin solubility, to reach 100 mg·kg^−1^·day^−1^ i.p was performed every 12 h with 50 mg/kg. Euthanasia was performed after the treatment and the animals received an i.p. BrdU injection (100 mg/kg) 1 h before euthanasia.

### 4.10. Tissue Processing, Histopathology, H&E, and Immunofluorescence

After euthanasia, the colons were removed, longitudinally opened, cleaned with phosphate-buffered saline (PBS) and fixed in 4% buffered paraformaldehyde for 24 h at 4 °C. The swiss-rolls were processed in sequential ethanol and xilol for paraplast inclusion and the tissues were sectioned in Leica RM2125 RTS microtome and stained with hematoxylin and eosin (H&E) for microscopic identification of lesions, adenomas, and adenocarcinomas. Indirect immunofluorescence was performed after serial deparaffinization in xilol and ethanol. Heat induced epitope retrieval was performed in sodium citrate buffer (10 mM Sodium citrate, 0.05% Tween 20, pH 6.0) in a steamer and nonspecific binding sites were blocked with bovine serum albumin 3% in PBS. The sections were incubated with monoclonal antibodies rat anti-Ki-67 (Invitrogen #14569882; 1:100) and mouse anti-BrdU (GE Healthcare #RPN202; 1:3) overnight at 4 °C. Anti-rat biotinylated and anti-mouse Alexa 488-conjugated secondary antibodies (Invitrogen) were used to visualize Ki-67 and BrdU, respectively and the nuclei were stained with DAPI. Images were captured using the Olympus Light Microscope BX53 with a LUCPLFLN 20XPH objective and a SC50 color camera (Olympus Life Science Solutions America, Waltham, MA, USA). Only the adenocarcinoma areas confirmed in H&E staining were considered in immunofluorescence analysis. All images were manually and independently counted by at least two authors (LFSO, JMAD and AF). The proportion of positive-stained nuclei in the epithelial crypt cells found in the adenocarcinoma area were calculated and compared between the groups.

### 4.11. Statistical Analysis

For MTT assays statistical analysis we used a two-way ANOVA following a Bonferroni post-test (GraphPad Prism version 6.0), error bars represent standard error. For all other results, we used a one-way ANOVA (GraphPad Prism version 6.0). Figures show the mean of three replicates performed three times; standard deviation and statistical significance was set at * *p* < 0.05 ** *p* < 0.01 *** *p* < 0.001.

## 5. Conclusions

In summary, our data describes lonchocarpin, a flavonoid from the chalcone class, as a potent inhibitor of the Wnt/β-catenin pathway that acts downstream to β-catenin stabilization level and impairs TCF mediated transcription. In addition, lonchocarpin presents anti-tumor growth properties in vitro and inhibits adenocarcinoma proliferation in an in vivo CRC model. Further studies should be conducted in order to improve its activity and perhaps propose lonchocarpin as an alternative in CRC treatment.

## Figures and Tables

**Figure 1 cancers-11-01968-f001:**
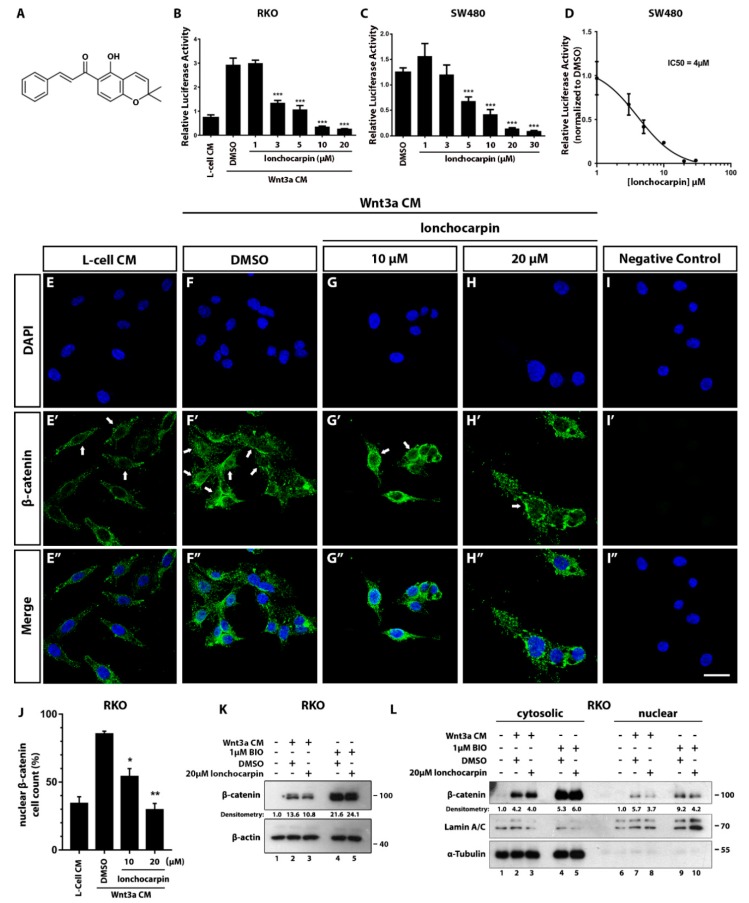
Lonchocarpin inhibits the Wnt/β-catenin pathway. Treatment with lonchocarpin inhibits Wnt reporter activity. (**A**) Lonchocarpin chemical structure, (**B**) RKO-pBAR/Renilla, and (**C**) SW480-pBAR/Renila gene reporter luciferase assay. (**D**) Lonchocarpin half maximal inhibitory concentration is 4 µM in the SW480-pBAR/Renilla cell lineage. Graph bars represent mean and SD. (**E**–**I**) SW480 β-catenin and DAPI immunocytochemistry staining showed that lonchocarpin decreases β-catenin translocation after lonchocarpin treatment. RKO cells were treated with (**E**–**E”**) L-cell control conditioned medium or Wnt3a conditioned medium with (**F**–**F”**) DMSO, (**G**–**G”**) 10 µM or (**H**–**H”**) 20 µM lonchocarpin. (**I**–**I”**) Immunocytochemistry negative control. (**J**) Quantification of nuclear β-catenin positive cell count ratio. Graph bars represent mean and SEM. (**K**) Immunoblotting depicting β-catenin and β-actin total levels of RKO cells treated with Wnt3a CM (conditioned medium) for 7 h. Densitometry is shown as the ratio of β-catenin/β-actin. (**K**) Immunoblot of cell lysate of RKO cells treated with 1-L-cell CM, 2-Wnt3a CM + DMSO, 3-Wnt3a CM + lonchocarpin 20 µM, 4-BIO 1 µM + DMSO, 5-BIO 1 µM + lonchocarpin 20 µM. (**L**) Immunoblot of cytosolic and nuclear fractions of RKO cells treated with 1,6-L-cell CM, 2,7-Wnt3a CM + DMSO, 3,8-Wnt3a CM + lonchocarpin 20 µM, 4,9-BIO 1 µM + DMSO, 5,10-BIO 1 µM + lonchocarpin 20 µM. Cytosolic densitometry was calculated considering α-Tubulin as the loading control, while nuclear densitometry considered Lamin A/C as the loading control. * *p* < 0.05, ** *p* < 0.01, *** *p* < 0.001. Scale bar represents 20 μm.

**Figure 2 cancers-11-01968-f002:**
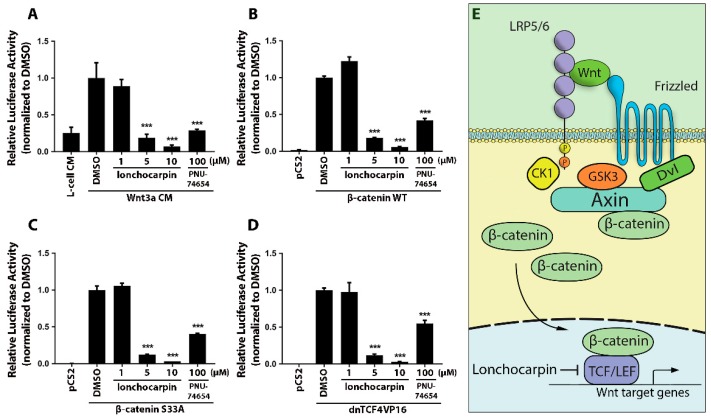
Lonchocarpin inhibits Wnt/β-catenin pathway downstream to TCF4. Lonchocarpin suppresses Wnt/β-catenin pathway induced by Wnt3a CM treatment (**A**) or by transfection with (**B**) β-catenin, (**C**) β-catenin S33A, or (**D**) dnTCF4VP16 in HEK293T cells. (**E**) Proposed lonchocarpin mechanism of action. Graph bars represent mean and SD. *** *p* < 0.001.

**Figure 3 cancers-11-01968-f003:**
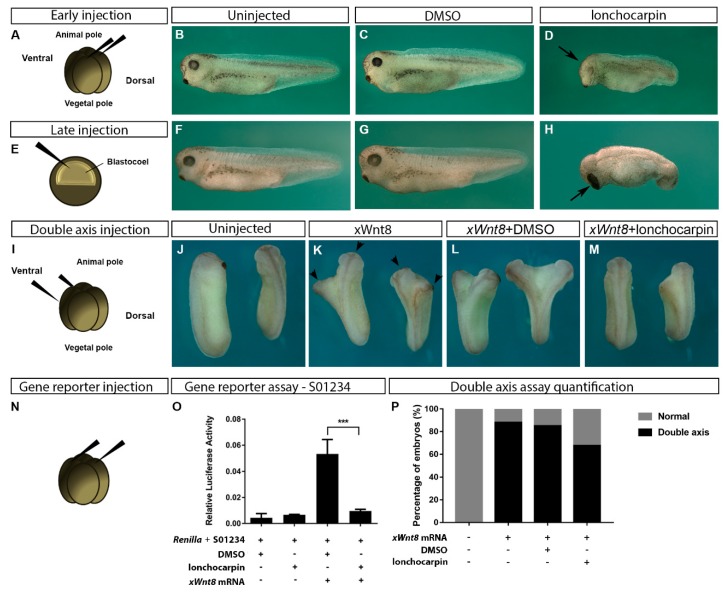
Lonchocarpin alters axial patterning in *Xenopus laevis* embryos and inhibits Wnt-8 induced axis and S01234-luciferase reporter. The 4-cell stage injected embryos display a smaller head compared uninjected and DMSO-injected embryos (**A**–**D**). Stage 9 blastulae injected embryos display a larger head (arrow) and cement gland (arrow) compared to uninjected or DMSO-injected embryos (**E**–**H**). Injection of xWnt8 mRNA into ventral blastomere induced ectopic axis (arrowhead) compared to uninjected embryos (**I**–**K**,**P**). Lonchocarpin inhibited Wnt8-induced secondary axis (**M**,**P**), but not DMSO (**L**,**P**). Lonchocarpin injection inhibits Wnt8-induced S01234-luciferase specific reporter activation in *Xenopus laevis* embryo (**N**,**O**) *** *p* < 0.001. Graph bars represent mean and SD (see also Appendix A).

**Figure 4 cancers-11-01968-f004:**
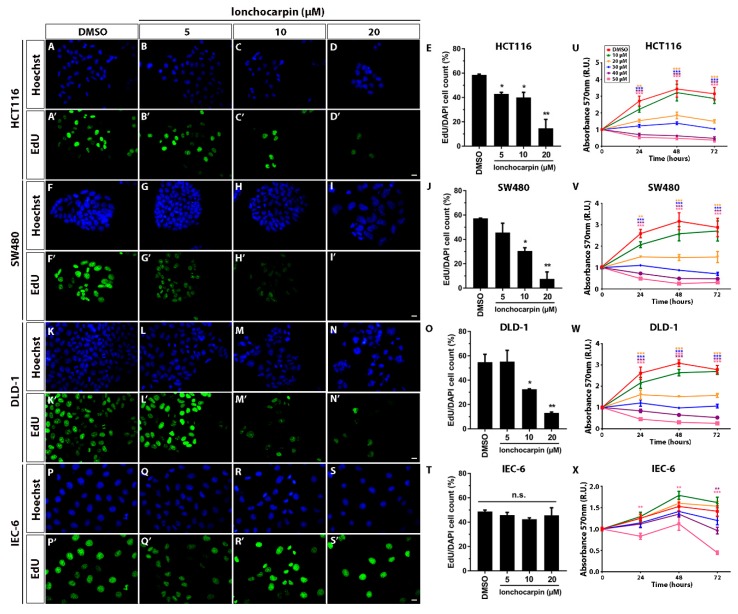
Lonchocarpin inhibits HCT116, SW480, and DLD-1 colorectal cancer cell lines proliferation. Proliferation assay shows that lonchocarpin suppresses proliferation of HCT116, SW480, and DLD-1 colorectal cancer cell lines, while not affecting IEC-6 non-tumor small intestine cell line proliferation. DAPI and EdU stained cells micrographs acquired 24 h post 5, 10, and 20 µM lonchocarpin treatment. (**A**–**E**) HCT116, (**F**–**J**) SW480, (**K**–**O**) DLD-1, (**P**–**T**) IEC-6. Graphs show the percentage of EdU positive cells. Scale bar represents 10 μm. Graph bars represent mean and SD. MTT assay shows that lower lonchocarpin concentrations are less cytotoxic to the non-tumor cell line IEC-6 compared to the tumor cell lines. MTT assay performed after treatment with 10, 20, 30, 40, and 50 µM lonchocarpin of (**U**) HCT116, (**V**) SW480, (**W**) DLD-1, and (**X**) IEC-6 cells during 24, 48, and 72 h. R.U. (Relative Units). * *p* < 0.1, ** *p* < 0.01. Graphs show mean and SEM.

**Figure 5 cancers-11-01968-f005:**
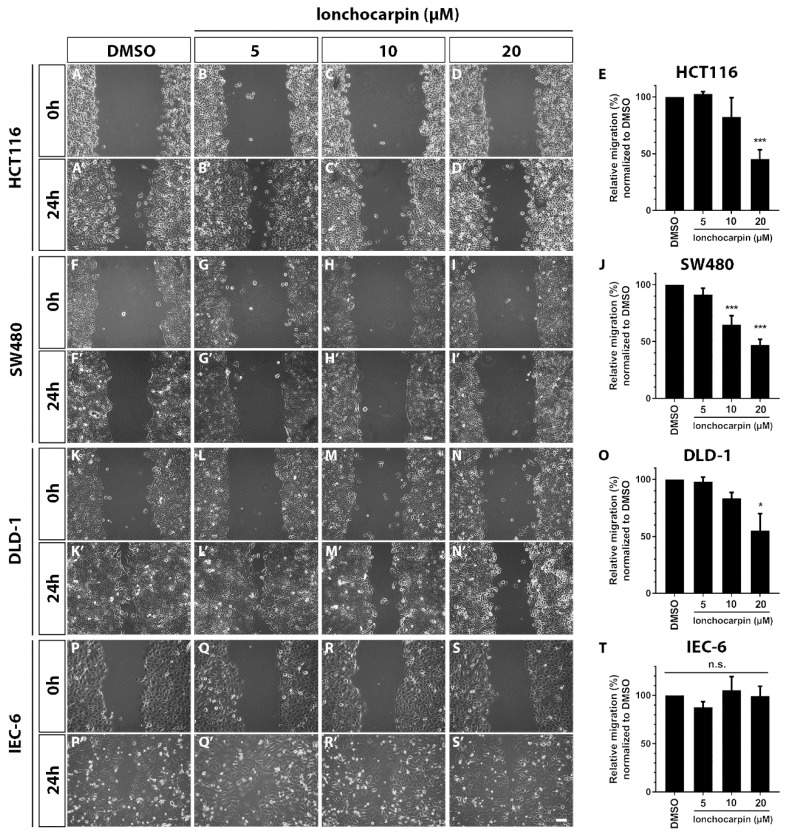
Lonchocarpin inhibits cell migration in HCT116, SW480, and DLD-1 colorectal cancer cell lines. Scratch assay shows that lonchocarpin impairs migration of HCT116, SW480, and DLD-1 colorectal cancer cell lines while not interfering with the IEC-6 non-tumor intestine cell line migration. Images show cell migration through the scratch healing 24 h post treatment with 5, 10, and 20 µM lonchocarpin (**A**–**E**) HCT116, (**F**–**J**) SW480, (**K**–**O**) DLD-1, (**P**–**T**) IEC-6. Graph shows relative wound area closure relative to time 0 h. All conditions were normalized to DMSO. * *p* < 0.1, *** *p* < 0.001. Scale bar represents 100 μm. Graph bars represent mean and SD.

**Figure 6 cancers-11-01968-f006:**
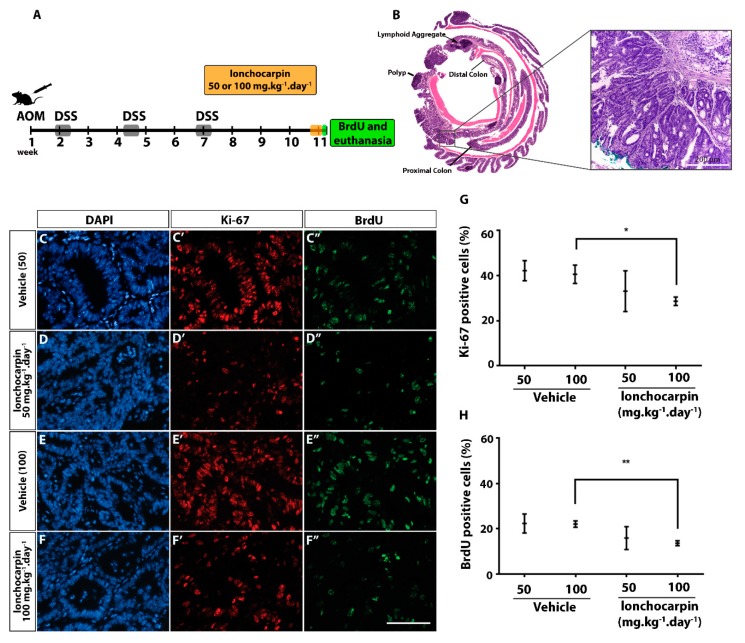
Lonchocarpin decreases cell proliferation in adenocarcinomas. (**A**) Timeline of inflammation-driven colon cancer tumorigenesis model (azoxymethane (AOM)/dextran sulfate sodium (DSS)) protocol in adult mice. Groups were divided as follows, vehicle 50 or 100: i.p. vehicle corresponding to volume for the respective dose in the treated group; treated groups: i.p. injections of 2.5 mg/mL lonchocarpin 50 or 100 mg·kg^–1^·day^–1^ for four days. Vehicle: PEG400 30% in sterile NaCl 0.9%. BrdU i.p. injection (100 µg/kg) 1 h before euthanasia. (**B**) Representative H&E swiss-roll image of a colon section from a mouse subjected to AOM/DSS protocol. The zoom picture shows the corresponding area of an adenocarcinoma. Scale bar represents 200 μm. (**C**) Representative immunofluorescence photomicrographs of adenocarcinoma areas from colon sections stained for proliferation markers (**C**’–**F**’) Ki-67 (red, Cy3) and (**C”**–**F**”) BrdU (green, Alexa 488). (**C**–**F**) Nuclei stained with DAPI (blue). (**D**,**E**) Quantification of the percentage of (**G**) Ki-67 (**H**) or BrdU positive cells per indicated group. Scale bar represents 100 μm. Graphs represent mean and SEM. * *p* < 0.05 ** *p* < 0.01 Student *t*-test of lonchocarpin treatment condition in comparison to vehicle.

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
