# Peer review of "The Chalcone Lonchocarpin Inhibits Wnt/β-Catenin Signaling and Suppresses Colorectal Cancer Proliferation"

_cancers, 2019, doi:10.3390/cancers11121968_

Round 1

Reviewer 1 Report

Dear Authors

  Thank you for the great work!

The results are convincing and interesting!

I have the following comments:

1) I wonder whether the dosage you use now can be applied to human use!!! If not, then the study will not have significance as the ultimate aim is to help the save the colorectal cancer patients.

2) Mouse models results are not very clear! See if you can provide more figures or tables summarizing them?

I look forward to the revised manuscript.

Author Response

Point 1. I wonder whether the dosage you use now can be applied to human use!!! If not, then the study will not have significance as the ultimate aim is to help the save the colorectal cancer patients.

We thank the reviewer for this discussion. It is well established in the literature concentrations between 25 to 100 mg/ kg for in vivo experiments with other natural compounds (Huo, Liu et al. 2016, Wu, Yu et al. 2018, Zhang, Su et al. 2018, Zhang, Wang et al. 2018, Zheng, Ma et al. 2018). A high dosage of lonchocarpin was used because the animals were exposed for only 4 days (acute treatment) when tumors were already stabilized.  To answer the question of whether lonchocarpin would be safe in the future for human treatment, we already started a follow-up project that envisions reducing the compound dosage and improving solubility by making modifications in the chemical structure of lonchocarpin and further enhancing its effects through nanodelivery systems. We believe this present manuscript is the first step to describe the lonchocarpin mechanistic model of action on Wnt signaling and the functional significance of impairing tumor growth. We propose as a next project to find the best mechanism to deliver lonchocarpin into the tumor tissue.

Huo, X., et al. (2016). "Flavonoids extracted from licorice prevents Colitis-associated carcinogenesis in AOM/DSS mouse model." International journal of molecular sciences 17(9): 1343.        

Wu, X., et al. (2018). "Radix Tetrastigma hemsleyani flavone exhibits antitumor activity in colorectal cancer via Wnt/β-catenin signaling pathway." OncoTargets and therapy 11: 6437.

Zhang, M.-J., et al. (2018). "Chemopreventive effect of Myricetin, a natural occurring compound, on colonic chronic inflammation and inflammation-driven tumorigenesis in mice." Biomedicine & Pharmacotherapy 97: 1131-1137.

Zhang, Y.-S., et al. (2018). "Natural dietary compound naringin prevents azoxymethane/dextran sodium sulfate-induced chronic colorectal inflammation and carcinogenesis in mice." Cancer biology & therapy 19(8): 735-744.

Zheng, R., et al. (2018). "Chemopreventive Effects of Silibinin on Colitis-Associated Tumorigenesis by Inhibiting IL-6/STAT3 Signaling Pathway." Mediators of inflammation 2018.

Point 2. Mouse models results are not very clear! See if you can provide more figures or tables summarizing them?

We thank the reviewer for this suggestion. We have added two tables summarizing the results (Table S1 and Table S2). We believe that the tables helped to clarify these data.

Reviewer 2 Report

Wnt cascade is one of the most important signal pathways in the cell, which is required for normal embryonic development, differentiation, maintenance of stem cell phenotype and migration. Mutations in this pathway are associated with tumor growth (especially with colon cancers, hepatocarcinomas and leukemias), where they participate in maintenance of tumor-initiating cells and metastasing. Because of that there is a considerable interest to develop inhibitors of Wnt pathway as anti-tumor agent. Preclinical studies have shown that inhibition of the Wnt signaling pathway leads to a therapeutic effect in a number of Wnt-dependent tumors. The development of antitumor compounds focuses on the inhibition of 4 parts of the cascade: palmitation of Wnt ligands by the protein porcupine inhibition of secretion), binding of Wnt ligands to its receptors and coreceptors, prevention of deactivation of the degrading complex and / or translocation of β-catenin to the nucleus and binding of β-catenin to other components of the transcriptional complex. There are currently 42 clinical trials linked to inhibition of the Wnt signaling pathway in the clinicalTrials.gov database.

Thus, this study is indeed relevant. I liked the article, it is structured, logically built, adequate methods were used and conclusions were correctly formulated.

I have a comment / wish to the authors of the article. It is known that the introduction of drugs into clinical practice is often limited by the high toxicity of drugs for normal tissues and the inadequacy of Wnt inhibition itself to inhibit tumor growth. Obviously, both the study of new targets in this signaling pathway and the selection of optimal combinations with existing antitumor agents are required. It seems to me in a discussion of the results it would be appropriate to compare the effectiveness of flavonoids and drugs undergoing clinical trials with respect to potential efficacy and toxicity. And also suggest that in combination with which chemotherapy drugs they can be effectively combined, if not, why.

Author Response

Point 1. I have a comment / wish to the authors of the article. It is known that the introduction of drugs into clinical practice is often limited by the high toxicity of drugs for normal tissues and the inadequacy of Wnt inhibition itself to inhibit tumor growth. Obviously, both the study of new targets in this signaling pathway and the selection of optimal combinations with existing antitumor agents are required. It seems to me in a discussion of the results it would be appropriate to compare the effectiveness of flavonoids and drugs undergoing clinical trials with respect to potential efficacy and toxicity. And also suggest that in combination with which chemotherapy drugs they can be effectively combined, if not, why.

We thank the reviewer for the suggestion, and we have added the discussion below to the manuscript. We believe that the discussion is now better structured and more enlightening.

“[40]. Together with Irinotecan, 5-FU is also widely used in the clinic. Intriguingly, Wnt signaling inhibition has been shown to decrease the resistance of colorectal cells to these chemotherapy drugs treatment [41]. Interestingly, MEK signaling inhibitors have been shown to increase canonical Wnt signaling in CRC [42], and the co-treatment of MEK inhibitors and Wnt signaling inhibitors resulted in a reduction of tumor growth [42]. Thus, MEK inhibitors should also be addressed as a cotreatment with lonchocarpin.”

“There are currently 55 clinical trials aiming to inhibit Wnt signaling pathway in cancers, in which 21 are CRC studies (clinicaltrials.gov). Among the 21 CRC clinical trials, only two tested compounds inhibit the canonical Wnt signaling at the transcriptional level, PRI-724 and resveratrol. In this context, in comparison with these compounds, the in vitro IC50 of lonchocarpin Wnt signaling inhibition is noticeable. PRI-724 inhibited the Wnt signaling pathway at 25 μM in vitro [43] (authors used cyclin D1 western blot to check Wnt signaling modulation), resveratrol inhibited at 20 μM [44] (authors used a Wnt signaling specific gene reporter assay), whereas 4 uM lonchocarpin reached the IC50. PRI-724 clinical trial has been withdrawn due to supply issues (NCT02413853) and resveratrol results have not been published yet (NCT00256334). We believe that lonchocarpin anticancer effects should be further addressed in preclinical studies, so it can be a possible clinical trial candidate clinical.”

Reviewer 3 Report

In this paper by Predes et al., authors have investigated and identified the chalcone lonchocarpin isolated from Lonchocarpus sericeus as a Wnt/β-catenin pathway inhibitor, both in vitro and in vivo. By means of appropriate methods they show that Lonchocarpin impairs β-catenin nuclear localization and also inhibits the constitutively active form of TCF4, dnTCF4-VP16, acting at the transcriptional level. This drug also displays selective activity showing inhibitory effects on cell migration and cell proliferation on three colorectal cancer cell lines, but not on the non-tumoral intestinal cells.

Since lonchocarpin reduces tumor proliferation on colorectal cancer mice model, Authors conclude that lonchocarpin is a novel Wnt/β-catenin inhibitor compound that impairs colorectal cancer cell growth in vitro and in vivo.

The work seems well conducted and is well written; the results obtained support the rationale and strengthen the conclusion, even because the natural compound they studied showed a more potent and efficient inhibitory effect than the previously published drug PNU-74654.

I have only minor concerns.

Figs 4 E, J and O report the effects of lonchocarpin and of the drug vehicle DMSO on the DAPI and EdU stained cells in the three cell lines. However, the bars of DMSO also are much lower than 100% that should be the level of untreated control cells; actually they are always around or below 60%.

This means that DMSO also showed an effect in this experiment???

Please explain better this result.

Again about Fig.4.

10 μM lonchocarpin showed activity in Figs 4 E, J and O, but not in the corresponding MTT experiments (4U to 4W), and also 20 μM lonchocarpin did not decrease relative 570nm absorbance.

Could this result indicate an interfering chemical reaction between MTT and lonchocarpin (Pagliacci et al., European Journal of Cancer, Volume 29, Issue 11, 1993)?? Or an action of lonchocarpin at mitochondrial level with the risk to under extimate the real effect??

Author Response

Point 1. Figs 4 E, J and O report the effects of lonchocarpin and of the drug vehicle DMSO on the DAPI and EdU stained cells in the three cell lines. However, the bars of DMSO also are much lower than 100% that should be the level of untreated control cells; actually they are always around or below 60%.

This means that DMSO also showed an effect in this experiment??? Please explain better this result.

Dear reviewer, thank you very much for this concern. Actually there was a mistake in the figure legend. The graph is not normalized to untreated condition, but shows the percentage of EdU positive cells (without normalization). So, in this case, DMSO is our control condition and due to unpublished data it does not affect cell proliferation. For these experiments DMSO was used at 0.1% which usually shows no effect.

Point 2. Again about Fig.4.

10 μM lonchocarpin showed activity in Figs 4 E, J and O, but not in the corresponding MTT experiments (4U to 4W), and also 20 μM lonchocarpin did not decrease relative 570nm absorbance.

Could this result indicate an interfering chemical reaction between MTT and lonchocarpin (Pagliacci et al., European Journal of Cancer, Volume 29, Issue 11, 1993)?? Or an action of lonchocarpin at mitochondrial level with the risk to under extimate the real effect??

We thank the reviewer for this discussion that is indeed very interesting. We used MTT assay intending to check cell viability, and not as a proliferation assay. However, if 10 μM lonchocarpin impairs proliferation we should see a decrease of 570nm absorbance at later time points. We believe the reviewer might be correct and there could be an interference of lonchocarpin treatment in the MTT assay.

Round 2

Reviewer 1 Report

Dear Authors

 The revised paper is very much improved now!

Congratulations and I will recommend "Acceptance"!

With Best Regards

Prof Wong